# Low-frequency transmission and persistence of antimicrobial-resistant bacteria and genes from livestock to agricultural soil and crops through compost application

**Akira Fukuda**[1], **Masato Suzuki**[2], **Kohei Makita**[3], **Masaru Usui**[1]*

**1** Food Microbiology and Food Safety Unit, Division of Preventive Veterinary Medicine, School of Veterinary Medicine, Rakuno Gakuen University, Ebetsu, Japan, **2** Department of Bacteriology II, National Institute of Infectious Diseases, Tokyo, Japan, **3** Veterinary Epidemiology Unit, Division of Preventive Veterinary Medicine, School of Veterinary Medicine, Rakuno Gakuen University, Ebetsu, Japan

* usuima@rakuno.ac.jp

**Data Availability Statement:** The assembled whole-genome sequences were deposited in a public database (DNA Data Bank of Japan).

## Abstract

Livestock excrement is composted and applied to agricultural soils. If composts contain antimicrobial-resistant bacteria (ARB), they may spread to the soil and contaminate cultivated crops. Therefore, we investigated the degree of transmission of ARB and related antimicrobial resistance genes (ARGs) and, as well as clonal transmission of ARB from livestock to soil and crops through composting. This study was conducted at Rakuno Gakuen University farm in Hokkaido, Japan. Samples of cattle feces, solid and liquid composts, agricultural soil, and crops were collected. The abundance of *Escherichia coli*, coliforms, *β*-lactam-resistant *E. coli*, and *β*-lactam-resistant coliforms, as well as the copy numbers of ARG (specifically the *bla* gene related to *β*-lactam-resistant bacteria), were assessed using qPCR through colony counts on CHROMagar ECC with or without ampicillin, respectively, 160 days after compost application. After the application of the compost to the soil, there was an initial increase in *E. coli* and coliform numbers, followed by a subsequent decrease over time. This trend was also observed in the copy numbers of the *bla* gene. In the soil, 5.0 CFU $g^{-1}$ *E. coli* was detected on day 0 (the day post-compost application), and then, *E. coli* was not quantified on 60 days post-application. Through phylogenetic analysis involving single nucleotide polymorphisms (SNPs) and using whole-genome sequencing, it was discovered that clonal $bla_{CTX-M}$-positive *E. coli* and $bla_{TEM}$-positive *Escherichia fergusonii* were present in cattle feces, liquid compost, and soil on day 0 as well as 7 days post-application. This showed that livestock-derived ARB were transmitted from compost to soil and persisted for at least 7 days in soil. These findings indicate a potential low-level transmission of livestock-associated bacteria to agricultural soil through composts was observed at low frequency, dissemination was detected. Therefore, decreasing ARB abundance during composting is important for public health.

Accession numbers are shown in S3 Table. All genome sequence files are available from the DDBJ database via accession number(s) SAMD00518796-00518813. The following links enable access to the DDBJ database and the provided accession numbers (SAMD00518796-00518813) can be utilized to retrieve the minimal data set associated with our manuscript: https://ddbj.nig.ac.jp/resource/biosample/SAMD00518796 https://ddbj.nig.ac.jp/resource/biosample/SAMD00518797 https://ddbj.nig.ac.jp/resource/biosample/SAMD00518798 https://ddbj.nig.ac.jp/resource/biosample/SAMD00518799 https://ddbj.nig.ac.jp/resource/biosample/SAMD00518800 https://ddbj.nig.ac.jp/resource/biosample/SAMD00518801 https://ddbj.nig.ac.jp/resource/biosample/SAMD00518804 https://ddbj.nig.ac.jp/resource/biosample/SAMD00518805 https://ddbj.nig.ac.jp/resource/biosample/SAMD00518802 https://ddbj.nig.ac.jp/resource/biosample/SAMD00518803 https://ddbj.nig.ac.jp/resource/biosample/SAMD00518808 https://ddbj.nig.ac.jp/resource/biosample/SAMD00518809 https://ddbj.nig.ac.jp/resource/biosample/SAMD00518811 https://ddbj.nig.ac.jp/resource/biosample/SAMD00518812 https://ddbj.nig.ac.jp/resource/biosample/SAMD00518813.

**Funding:** This study was supported by a grant from the Food Safety Commission, Cabinet Office, Government of Japan (Research Program for Risk Assessment Study on Food Safety; No: JPCAFSC20202002), and partially supported by JSPS KAKENHI (Grant Numbers 19H04285 and 23H03553).

**Competing interests:** The authors have declared that no competing interests exist.

## Introduction

Livestock excrement is processed via composting and is used to promote the growth of crops [1]. Various composting methods, including aerobic composting and anaerobic digestion, are employed to guarantee the eradication of bacteria existing in excrements. Nevertheless, achieving total bacterial elimination from livestock excrement through composting proves to be challenging. Residual bacteria in composts, encompassing antimicrobial-resistant bacteria (ARB) and pathogens, pose a potential risk of spreading to agricultural soils [2,3]. In addition, bacteria in composts applied to soils stay in the soil and flow to surrounding environments [4]. Therefore, applying composts, especially via inadequate composting, to soils is a risk factor for disseminating livestock-associated bacteria, including ARB.

Compost application amplifies the presence of livestock-associated ARB, which are bacteria remaining in composts derived from livestock excrements in soils [5]. Additionally, compost harbors diverse remnants of antimicrobials utilized in livestock raising. These remnants can exert selective pressure on ARB, contributing to their resilience against these residual antimicrobials [6,7]. These studies were mainly employed genome-based analyses such as quantitative PCR to quantify various types of antimicrobial resistance genes (ARGs) as well as 16S amplification sequencing via short-read next-generation sequencing to unveil the microbiota [2,8]. Bacteria in the soil could contaminate crops and may transmit livestock-associated bacteria to humans [9]. The persistence and transmission of specific livestock-associated ARB from livestock to agricultural fields through composts are unclear.

In agricultural soils, fecal bacteria maintenance is influenced by various factors, such as temperature, water content, and nutrients [10]. Through the application of compost to soils, pathogenic bacteria derived from the composts are sustained within agricultural soils for a month [9,11]. Thus, quantitative analysis over time is needed to assess the risks of bacterial dissemination through composts.

$\beta$-lactams are used in livestock and humans, and cephalosporins are especially important antimicrobial agents in human clinical settings [12]. Resistance to $\beta$-lactams is mainly related to the presence of the *bla* gene and the production of $\beta$-lactamase in gram-negative bacteria [13]. ARGs can be transferred and disseminated across genera, particularly those mediated by mobile genetic elements [14]. Plasmid-mediated ARGs from bacteria in the compost can be transferred to bacteria in the soil [15]. Therefore, *bla* genes are among the important ARGs and should be investigated to determine the risks of dissemination. To control ARB, the transmission and dynamics of ARGs must be investigated.

If composting is inadequate, residuals containing ARB and ARGs derived from livestock could spread and transmit to soils and crops. This study aimed to investigate the quantitative and qualitative transmission of livestock-derived ARB and ARGs in compost to soils and crops. First, ARB and ARG abundance was determined in cattle feces, composts, agricultural soils, and crops in the same field to evaluate the influence of compost application to agricultural soils. Second, ARB were isolated from cattle feces, composts, agricultural soils, and crops in the same field to clarify the transmission of clonal ARB isolates from livestock to soils through composts.

## Materials and methods

### Sampling locations and sample collection

The samples used in this study were collected from the Rakuno Gakuen University farm in Hokkaido, Japan. This encompassed a field spanning 51 000 m$^2$ dedicated to the cultivation of corn. On the field, composts were applied once a year. In the Rakuno Gakuen University farm,

approximately 180 dairy cattle were kept, and their excretions were treated via aerobic composting (solid compost) and anaerobic digestion (liquid compost) using a biogas plant. In aerobic composting, the temperature of the solid compost is raised by self-heating. For making a solid compost, 20% dairy cattle excretions, 50% bedding, and 30% feed residues and waste were mixed and stored at a covered, outdoor area. This mixture was turned over every 2 weeks and allowed to stand for 5 months. In anaerobic biogas digestion, liquid compost is treated at a constant temperature, removing most pathogens [16]. For making a liquid compost, 40% dairy cattle excretions, 40 bedding, and 20% farm effluent water were mixed and stored in biogas plant's tank. In the tank, the mixture was heated to promote fermentation and kept in a month. Solid and liquid composts are kept within their designated areas on the farm. This storage continues until they are ready for use as fertilizer in the soil where corn is cultivated.

All samples were collected using sterilized bottles and underwent processing on the same collection day using the following methods. Six cattle feces, six solid compost, one liquid compost, and six soil samples were collected before compost application. The six dairy cattle feces sample were collected from six individuals on the same farm. The six solid compost samples were obtained from six different parts of the compost pile, and one liquid compost sample was obtained from a digested liquid storage tank of the biogas plant. The six soil samples were collected from six different points where corn is cultivated.

On day 0, the solid ($3.76$ kg/m$^2$) and liquid ($2.0$ kg/m$^2$) composts derived from livestock excrements in the Rakuno Gakuen University farm were applied to the cultivated field. The grain corn was sown in the field on day 28 and harvested on day 160. After applying compost, soil (0, 7, 28, 60, 100, and 160 days), and corn (100 and 160 days), samples were collected from six predetermined locations, the same as those for pre-sample collection. Corn was divided into upper (leaves on day 100 and edible parts on day 160) and lower parts (roots) (S1 Table).

## Quantification and isolation of bacteria

Samples (2.5 g) were homogenized in 22.5 mL of buffered peptone water (Becton, Dickinson and Company, Franklin Lakes, NJ, USA). The homogenates were plated on CHROMagar™ ECC agar (CHROMagar, Paris, France) without and with 100 μg mL$^{-1}$ ampicillin (Sigma-Aldrich, St. Louis, MO, USA) to determine the abundance (CFU g$^{-1}$) of *Escherichia coli* and coliforms (excluding *E. coli*), as well as *β*-lactam-resistant *E. coli* and coliforms (excluding *E. coli*), respectively [17]. After incubation at 37 ˚C for 20 h, *E. coli* and coliform colonies were counted according to their colony colors (*E. coli*, blue colony; other coliforms, red colony) and morphologies. Up to three *E. coli* or coliform colonies were isolated from each agar plate. Means and standard errors were calculated for each type of sample. Additionally, to isolate other *Escherichia* species from samples wherein *E. coli* colonies did not grow, precultures with 2.5 and 1 g of samples were inoculated in 22.5 mL of Luria-Bertani broth (Invitrogen, Carlsbad, CA, USA) and Colilert-18 (IDEXX, Westbrook, ME, USA), respectively, and incubated at 37 ˚C for 20 h. Then, the preculture was streaked onto CHROMagar ECC agar without and with 100 μg mL$^{-1}$ ampicillin and incubated at 37 ˚C for 20 h. Up to three *E. coli* or coliform colonies were isolated from each agar plate.

## Characterization of bacteria

For each isolate, the bacterial species were determined using matrix-assisted laser desorption/ionization-time of flight mass spectrometry (MALDI-TOF MS) using a Bruker MALDI Biotyper system (Bruker Daltonics, Bremen, Germany) [18]. In isolates identified as *Escherichia* genus members using MALDI-TOF MS, biochemical tests and PCR were conducted to identify the *Escherichia* species [19].

For Enterobacterales, the minimum inhibitory concentrations were determined according to the Clinical Laboratory Standards Institute (CLSI) guidelines [20]. In addition, the susceptibility of the isolates to ampicillin, cefazolin, cefotaxime, tetracycline, nalidixic acid, ciprofloxacin, and gentamicin was evaluated using the agar dilution method. All antimicrobials were obtained from Sigma-Aldrich. Resistance breakpoints were defined according to the CLSI guidelines [20]. *E. coli* ATCC25922 and *Pseudomonas aeruginosa* ATCC27853 were used as quality control strains.

For $\beta$-lactam (ampicillin, cefazolin, or cefotaxime)-resistant isolates, $bla_{TEM}$, $bla_{CTX-M}$, $bla_{SHV}$, and AmpC $\beta$-lactamase genes were detected using PCR [21].

When more than two isolates from an agar plate exhibited the same antimicrobial susceptibility profile and possessed ARGs, they were considered single isolates.

## DNA extraction and quantitative PCR (qPCR)

To quantify the genes in each field sample, DNA was extracted from 0.2 g of cattle feces, solid compost, and liquid compost samples using ISOFECAL Kit (Nippon Gene, Tokyo, Japan) and from soil and corn samples using ISOIL Kit (Nippon Gene), according to the manufacturer's instructions. qPCR was then performed to determine the copy numbers of *bla* ($bla_{TEM}$, $bla_{CTX-M}$, and $bla_{SHV}$) and *uidA* genes (marker gene of *E. coli*) using TB Green® Premix Ex Taq™ II (TaKaRa, Shiga, Japan) in 20 μL reactions containing 2.5 μL of template DNA and 0.4 μM of each primer (S2 Table) [22,23]. The reaction conditions were initial denaturation at 95 ˚C for 30 s, followed by 45 cycles each at 95 ˚C for 5 s, and the optimal melting temperature of each gene for 30 s. Melting curves were analyzed to verify the absence of nonspecific amplifications. All reactions included negative and positive controls. Samples showing results below the limit of detection were considered negative. The standard curves were generated by cloning the amplicon from the genes into the pTA2 vector and transforming it into *E. coli* DH5α competent cells [7]. Means and standard errors were calculated for each type of sample.

## Whole genome sequencing

The 18 *bla*-positive isolates of $\beta$-lactam-resistant strains were subjected to whole-genome sequencing. Genomic DNA for short-read sequencing was extracted using a QIAquick®PCR Purification Kit (QIAGEN, Hilden, Germany). Libraries were prepared using a Nextera XT DNA Library Prep Kit (Illumina, San Diego, CA, USA). Multiplexed paired-end sequencing was performed using HiSeq X (2 × 150 bp paired-end; Illumina). Reads of the adaptor sequences were trimmed and subsequently assembled *de novo* into contigs using Shovill v1.1.0 (https://github.com/tseemann/shovill) with the default parameters. ARGs were identified using the ResFinder database on the Center for Genomic Epidemiology server (http://www.genomicepidemiology.org/). CSI phylogeny v1.4 (available online at https://cge.food.dtu.dk/services/CSIPhylogeny/) was used to construct the SNP-based phylogeny [24]. CSI Phylogeny v1.4 used BWA version 0.7.12, SAMtools version 0.1.18, BEDtools version 2.16.2, MUMmer version 3.23, and FastTree version 2.1.7. *E. coli* MG1655 was used as the reference genome.

## Statistical analysis

Statistical significance was determined using the Mann–Whitney U test, with the significance threshold set at $p < 0.05$.

## Results

### Bacterial abundance

Bacterial abundance with average numbers is shown in Fig 1 (S3 Table). No significant difference in bacterial abundance was detected in agricultural soils after compost application. After compost application to agricultural soils, 5.0, 4.0, 4.0, and 3.4 CFU g$^{-1}$ *E. coli* were detected in the soil (days 0, 7, 28, and 160, respectively) (Fig 1A). In cattle feces and liquid compost, $2.5 \times 10^4$ and $2.0 \times 10^2$ CFU g$^{-1}$ *E. coli* were quantified, respectively. *E. coli* was not quantified in the soil (pre and days 60 and 100), the corn (days 100 and 160), or solid compost. In addition, $3.3 \times 10^2$–$7.5 \times 10^3$ and $6.4 \times 10^3$–$3.2 \times 10^6$ CFU g$^{-1}$ coliforms, excluding *E. coli*, were quantified in soils and corn, respectively. In cattle feces, solid compost, and liquid compost, $5.9 \times 10^2$, $3.5 \times 10^3$, and 5.2 CFU g$^{-1}$ coliforms, excluding *E. coli*, were detected, respectively.

For *β*-lactam-resistant *E. coli*, there were 2.4 and $8.9 \times 10^1$ CFU g$^{-1}$ in the soil (day 0) and cattle feces, respectively, and were not quantified in other samples (Fig 1B). In addition, 1.2–$8.9 \times 10^1$ and $1.7$–$5.3 \times 10^1$ CFU g$^{-1}$ *β*-lactam-resistant coliforms, excluding *E. coli*, were quantified in soils and corn, respectively. In cattle feces and solid compost, $4.4 \times 10^1$ and $2.4 \times 10^1$ CFU g$^{-1}$ *β*-lactam-resistant coliforms, excluding *E. coli*, were quantified, respectively. In contrast, any *β*-lactam-resistant coliforms, excluding *E. coli*, were not quantified in liquid compost.

### Copy numbers of *bla* and *uidA* genes

The quantification results for the *bla* and *uidA* genes with average numbers are shown in Fig 2 (S4 Table). The *bla*$_{\text{SHV}}$ gene could not be detected, except in solid composts (3.7 copies g$^{-1}$). In soils (pre), 7.4 copies g$^{-1}$ of *bla*$_{\text{CTX-M}}$ and 3.0 copies g$^{-1}$ of *uidA* genes were quantified, and *bla*$_{\text{SHV}}$ and *bla*$_{\text{TEM}}$ genes were not quantified. After compost application, $3.6 \times 10^2$ copies g$^{-1}$

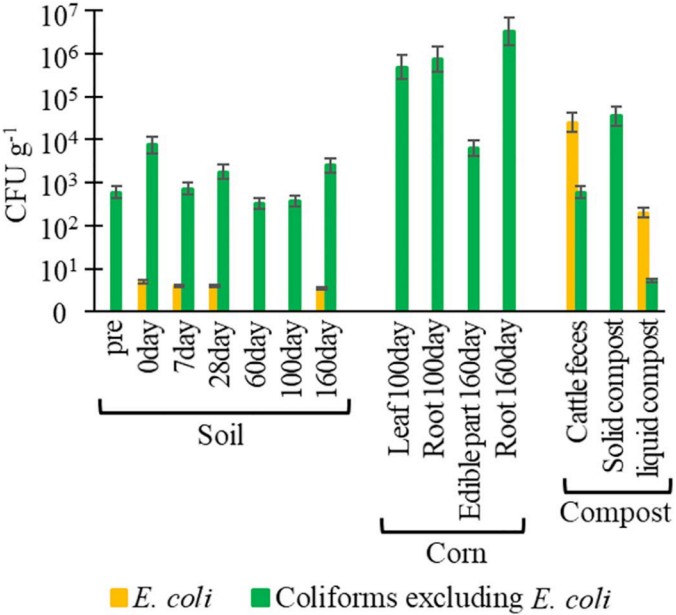

**Fig 1. Abundance of (A) *Escherichia coli* and coliforms (excluding *E. coli*), and (B) *β*-lactam-resistant *E. coli* and *β*-lactam-resistant coliforms (excluding *E. coli*) in soil, corn, and compost.** Pre: before the application of composts, day 0: day of application of compost to soils.

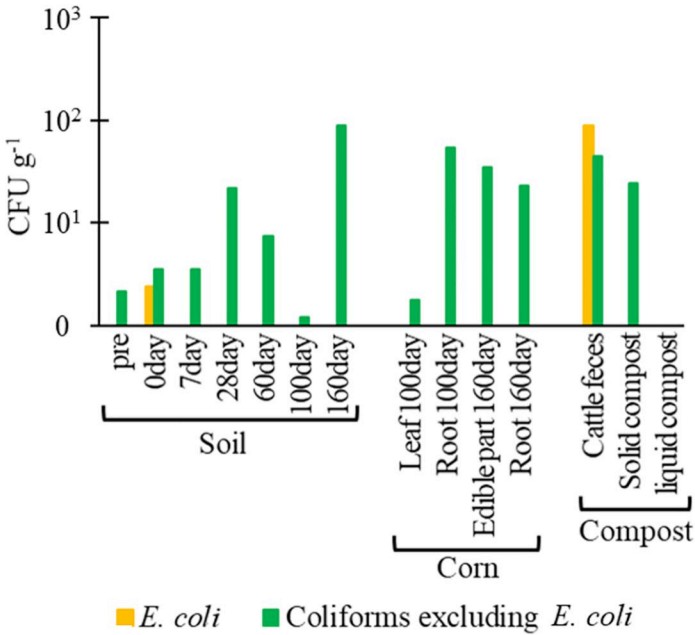

**Fig 2. Quantification of *bla* and *uidA* genes in soil, corn, and compost via qPCR.**

of $bla_{TEM}$, $1.8 \times 10^3$ copies g$^{-1}$ of $bla_{CTX-M}$, and $2.0 \times 10^2$ copies g$^{-1}$ of *uidA* genes were detected in soils (0 days). Further, copy numbers of these genes in soil (0 day) were significantly increased compared to those in soil (pre) ($p < 0.05$). From days 7 to 100, $5.7$–$4.4 \times 10^2$ copies g$^{-1}$ of $bla_{TEM}$, $2.7 \times 10^2$–$5.1 \times 10^2$ copies g$^{-1}$ of $bla_{CTX-M}$, and $2.6 \times 10^2$–$9.9 \times 10^2$ copies g$^{-1}$ of *uidA* genes were detected in soils. In soil (day 160), 3.8 copies g$^{-1}$ of $bla_{TEM}$, $2.3 \times 10^2$ copies g$^{-1}$ of $bla_{CTX-M}$, and $3.0 \times 10^1$ copies g$^{-1}$ of *uidA* genes were detected, and the copy number of $bla_{TEM}$ in soil (day 160) wassignificantly decreased compared to that in soil (day 0) ($p < 0.05$). In leaves of corn, 3.9 copies g$^{-1}$ of $bla_{TEM}$ and 3.3 copies g$^{-1}$ of *uidA* genes were quantified, and the $bla_{CTX-M}$ gene was not quantified. In the root of corn, $2.4 \times 10^1$ and 3.8 copies g$^{-1}$ of $bla_{TEM}$, $6.8 \times 10^1$ and 3.1 copies g$^{-1}$ of $bla_{CTX-M}$, and $3.6 \times 10^1$ and 2.9 copies g$^{-1}$ of *uidA* genes were detected on days 100 and 160, respectively. In edible parts of corn, $bla_{TEM}$, $bla_{CTX-M}$, and *uidA* genes were detected. In cattle feces, solid composts, and liquid compost, we detected $1.3 \times 10^4$, $1.0 \times 10^4$, and $1.8 \times 10^4$ copies g$^{-1}$ of the $bla_{TEM,}$ $1.1 \times 10^4$, $3.5 \times 10^4$, and $5.7 \times 10^2$ copies g$^{-1}$ of $bla_{CTX-M}$ and $1.1 \times 10^6$, 3.2, and $1.1 \times 10^4$ copies g$^{-1}$ of *uidA*, respectively.

### *Escherichia* isolation and characterization of *bla*-positive isolates

*E. coli* was isolated from all samples using a pre-culture step, excluding the edible parts of corn (day 160) and solid composts. $\beta$-lactam-resistant *E. coli* was isolated from soil (days 0 and 7), cattle feces, and liquid compost, while $\beta$-lactam-resistant *Escherichia fergusonii* was isolated from soils (pre and day 0), cattle feces, and liquid composts.

Ten *bla*-positive *E. coli* and eight *bla*-positive *E. fergusonii* were detected in $\beta$-lactam-resistant isolates from soils (pre and, days 0 and 7), cattle feces, and liquid compost (S5 Table). The *bla* genes were not detected in $\beta$-lactam-resistant isolates from soil (days 28, 60, 100, and 160), corn, and solid composts. From soils (days 0 and 7), cattle feces, and liquid compost, seven $bla_{CTX-M-2}$- and three $bla_{TEM}$-positive *E. coli* isolates were detected. From soils (pre and day 0),

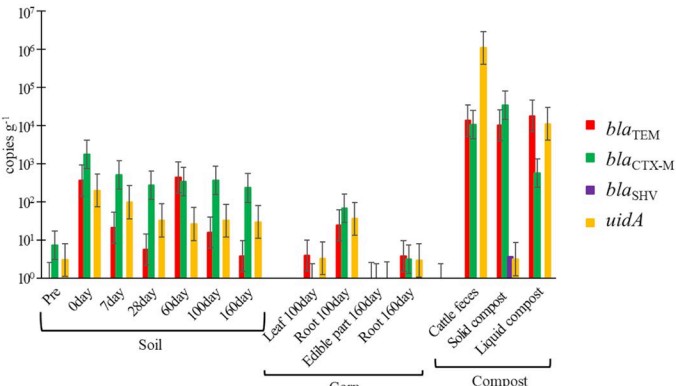

**Fig 3. Phylogenetic tree of *bla*-positive *β*-lactam-resistant *Escherichia* isolates from soil, cattle feces, and compost.** The phylogenetic tree was inferred from CSI phylogeny using the assembled contigs. The nodes show the strain №/source.

cattle feces, and liquid composts, eight *bla*$_{TEM}$-positive *E. fergusonii* isolates were detected. In addition to harboring *β*-lactam resistance and *bla* genes, these isolates were resistant to other antimicrobials that we tested and carried ARGs, except for two *bla*$_{TEM}$-positive *E. coli* isolates.

The phylogenetic tree of *bla*-positive isolates is shown in Fig 3. Closely related isolates which were considered clonal isolates, were observed by SNP-based phylogenetic analysis in seven *bla*$_{CTX-M-2}$-positive *E. coli* (F2-4, S11-4E, S13-4E, T1-4E, F4-4, F3-4E, and F1-4) isolated from soils (0 and 7 days), cattle feces, and liquid compost and in six *bla*$_{TEM}$-positive *E. fergusonii* (S7-4, S10-5E, F3-4, F1-7, T1-4, and F2-7) isolated from soils (pre and day 0), cattle feces, and liquid composts. Clonal isolates of seven *bla*$_{CTX-M-2}$-positive *E. coli* were resistant to ampicillin, cefazolin, and cefotaxime and possessed *aadA2*, *dfrA12*, and *sul1* genes. In addition, the seven clonal *bla*$_{CTX-M-2}$-positive *E. coli* were resistant to ampicillin, cefazolin, and cefotaxime and possessed *aadA2*, *dfrA12*, and *sul1* genes. In contrast, the six clonal *bla*$_{TEM}$-positive *E. fergusonii* were resistant to ampicillin and tetracycline and possessed *tetA*, *aph(3")-Ib*, *aph(6)-Id*, *dfrA14*, and *sul2* genes (S6 Table).

## Discussion

This study detected clonal *bla*-positive *Escherichia* species in cattle feces, liquid compost, and soil after compost application. These strains persisted for at least seven days in the soil, showing ARB transmission from livestock to agricultural soils would occur by applying processed composts in the tested field. Depending on the conditions, the abundance of *E. coli* decreased to the detection limit within 46–49 days at most in the agricultural fields [25]. After the application of cattle compost to soils, antimicrobial-resistant fecal coliforms experienced a rapid reduction within a week and reached the detection limit within 2 months [26]. After the application of chicken litter, closely related antimicrobial-resistant phenotypes of *Enterococcus* isolates were detected in the chicken litter and litter-amended soils [27]. These studies showed that livestock-associated bacteria, including ARB, could be transmitted to soils through processed composts and maintained in the soil for a certain period.

In this study, the number of *β*-lactam-resistant *E. coli* and coliform and copies of ARGs and *uidA* decreased with excrement processing. However, in the soil, it once increased immediately after applying compost especially the copy numbers of ARGs and *uidA* genes. After waste treatment processing, the microbiota changed. However, some enriched bacteria could harbor

ARGs and maintain their abundance in the microbiome [7]. Previous reports showed that ARB and ARGs decreased with appropriate composting [2,28] and increased after the composts were applied to the agricultural soils [18,29]. Although the microbiota structures of feces, composts, and soils were different, compost application in agricultural soils did not impact the overall structure of the microbiota [30,31]. Agricultural soils would be a suitable environment for fecal bacteria [25]. In the absence of antimicrobial selective pressure, it is difficult for ARB to maintain dominance among the total microbes of an environment because plasmids carrying ARGs require fitness costs and this causes inferior growth of ARB compared with that of the wild-type strains [30]. These studies suggest that appropriate livestock waste treatment for reducing ARB and ARGs as much as possible is important to prevent the spread of ARB and ARGs from livestock. Therefore, additional composting treatment strategies were needed to improve the removal of ARB and ARGs.

In the edible part of corn, *E. coli* was not isolated, and the *bla* and *uidA* genes were not detected. Although crops could be contaminated by pathogenic *E. coli* suspected to be livestock-associated from soil [32,33], the phyllosphere, endosphere, and soil microbiota were quite different [34,35], and the degree of bacterial transmission and persistence from soil to crops would vary according to the type of vegetable [36]. Another study showed that ARGs were transmitted into plants mainly through endophytic bacteria [37]. In crop production, many factors are involved in bacterial contamination, and contamination through soil could be a risk factor [27,38]. In our study, bacterial contamination from soil to crops was not observed, which could be attributed to the low abundance of targeted bacteria in the soil. Careful washing, such as rinsing with running water, is important for decreasing the abundance of bacteria, including ARB, and pathogens in the soil and preventing crop contamination at production and cooking sites [39,40].

After compost application, ARGs and *uidA* gene expression increased in the soil, suggesting the dissemination of ARGs and fecal bacteria derived from livestock excretions in the soil [8]. Additionally, certain amounts of ARGs were detected in the soil for up to 160 days. Through the treatment process of excrement, antimicrobial resistance and livestock-associated bacterial genes were reduced. However, compared with bacteria, including ARB, they were maintained for a longer time and not markedly reduced [7,28,41]. This result is because ARGs were transferred to bacteria across genera and maintained in bacteria that were not detected through cultivation methods in the experiments [15,42]. Another reason is that genes in dead cells and extracellular DNA of bacteria have been detected via qPCR and metagenomic analysis [43,44], leading to a higher risk estimate, making assessing the actual extent of ARB and ARG dissemination impossible. Therefore, the actual existence of functional antimicrobial resistance genes needs to be accurately assessed by measuring DNA only from living bacteria cells, such as by qPCR using monoazide. This process must be done to develop effective treatment methods for livestock excrement and prevent the dissemination of livestock-associated ARGs [43,44].

Clonal antimicrobial-resistant *E. fergusonii* was isolated from feces, liquid compost, and soil. Antimicrobial-resistant *E. fergusonii* from a different genetic background was isolated from the soil before compost application. Recently, *E. fergusonii* was detected in livestock and has attracted attention as a reservoir of ARGs [45]. Unlike *E. coli*, *E. fergusonii* is negative for lactose, sucrose, and sorbitol fermentation [46]. Because lactose and/or sucrose-positive fermentation was set to indicate Enterobacteriaceae (including *E. coli*) in isolation steps using agars, such as deoxycholate-hydrogen sulfide-lactose and MacConkey agars, *E. fergusonii* was overlooked in the bacterial isolation steps. In some cases, *E. fergusonii* caused human infections [19] and resisted last-line antimicrobials for human infections [47,48]. These findings suggest that *E. fergusonii*, which has not received much attention until recently, is an ARG

reservoir. Further investigation of overlooked bacteria, such as *E. fergusonii*, is important to prevent ARB transmission and maintenance.

## Conclusions

Compost application to the agricultural soil led to the introduction of compost-derived bacteria, but these bacteria exhibited low relative abundance, and the compost treatment exhibited no marked impact on the overall composition of the microbiome and resistome [49]. The notion that the extent of ARB spread from livestock to crops is modest, while the environmental dissemination remains evident. Therefore, the extent of removal of livestock-derived microorganisms from vegetables via customary washing processes and effective removal methods warrants further investigation.

## Supporting information

**S1 Table. Sample collection.**
(XLSX)

**S2 Table. Primers for quantitative PCR.**
(XLSX)

**S3 Table. Number of colonies forming unit.**
(XLSX)

**S4 Table. Copies number of genes by qPCR.**
(XLSX)

**S5 Table. Characterization of *bla*-positive *β*-lactam-resistant *Escherichia*.**
(XLSX)

**S6 Table. Sequence statistics of whole-genome sequence of *bla*-positive *β*-lactam-resistant *Escherichia*.**
(XLSX)

## Acknowledgments

We thank the staff of the Rakuno Gakuen University farm, Ebetsu City, Hokkaido, Japan.

## Author Contributions

**Conceptualization:** Akira Fukuda, Masaru Usui.

**Data curation:** Akira Fukuda, Masato Suzuki, Kohei Makita, Masaru Usui.

**Formal analysis:** Akira Fukuda, Masato Suzuki, Masaru Usui.

**Funding acquisition:** Masaru Usui.

**Investigation:** Akira Fukuda, Masaru Usui.

**Methodology:** Akira Fukuda, Masato Suzuki, Masaru Usui.

**Validation:** Akira Fukuda, Masaru Usui.

**Visualization:** Akira Fukuda.

**Writing – original draft:** Akira Fukuda, Masaru Usui.

**Writing – review & editing:** Akira Fukuda, Masato Suzuki, Kohei Makita, Masaru Usui.

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
