## [Decision Letter · Decision Letter 0]

4 Dec 2023

PONE-D-23-31188Low-Frequency Transmission and Persistence of Antimicrobial-Resistant Bacteria and Genes from Livestock to Agricultural Soil and Crops through Compost ApplicationPLOS ONE

Dear Dr. Usui,

Thank you for submitting your manuscript to PLOS ONE. After careful consideration, we feel that it has merit but does not fully meet PLOS ONE’s publication criteria as it currently stands. Therefore, we invite you to submit a revised version of the manuscript that addresses the points raised during the review process.

We look forward to receiving your revised manuscript.

Kind regards,

Gabriel Trueba, PhD

Academic Editor

PLOS ONE

Journal Requirements:

"This study was supported by a grant from the Food Safety Commission, Cabinet Office, Government of Japan (Research Program for Risk Assessment Study on Food Safety; No: JPCAFSC20202002), and partially supported by JSPS KAKENHI (Grant Numbers 19H04285 and 23H03553)."

5. Thank you for stating in your Funding Statement: 

"This study was supported by a grant from the Food Safety Commission, Cabinet Office, Government of Japan (Research Program for Risk Assessment Study on Food Safety; No: JPCAFSC20202002), and partially supported by JSPS KAKENHI (Grant Numbers 19H04285 and 23H03553)."

Reviewers' comments:

Reviewer's Responses to Questions

**Comments to the Author**

1. Is the manuscript technically sound, and do the data support the conclusions?

Reviewer #1: No

Reviewer #2: Partly

2. Has the statistical analysis been performed appropriately and rigorously? 

Reviewer #1: No

Reviewer #2: N/A

3. Have the authors made all data underlying the findings in their manuscript fully available?

Reviewer #1: No

Reviewer #2: Yes

4. Is the manuscript presented in an intelligible fashion and written in standard English?

Reviewer #1: Yes

Reviewer #2: Yes

5. Review Comments to the Author

Reviewer #1: 1. There are many errors from the viewpoint of science, for example, the description of HGT on Line 60-61, the use of 100 ug/mL for ampicillin resistant experiment on Line 107.

2. The objective of this work should be more clear elaborated. For example, bla genes as the valid ARG indicator, is it fully convincable?

3. As to the isolation of bacteria, how to differentiate by their colony colors and what does it mean "from samples showing zero E.coli count" on Line 104?

4. The biggest problem for qPCR is ambitious. Does the sample of isolates (Line 122-123) or feces/compost/soil (Line 128-129) all used? It is rather confusing.

5. What does it mean on Line 189-190, Line 195 "bla-positie b-lactam-resistant"?

6. The section of conclusion didn't incorporate with the experimental results and data in this work. It should be improved a lot.

Reviewer #2: On account of the manuscript PONE-D-23-31188, entitled “Low-Frequency Transmission and Persistence of Antimicrobial-Resistant Bacteria and Genes from Livestock to Agricultural Soil and Crops through Compost Application” by Akira Fukuda et al., the authors evaluated the quantitative and qualitative transmission of livestock-derived antimicrobial resistant bacteria (ARB) and antimicrobial resistance genes (ARGs) from livestock to soil and crops through composting in Japan. The topic is important to conduct human health risk assessment for antimicrobial resistance (AMR) in the agricultural environment. After careful consideration, I feel that this manuscript is to be published after improvement of some major shortcomings. Details of my comments are as follows:

1) The view point of this research is interesting, and the authors got interesting results. Several important revisions are, however, required before publication. The first one is novelty of the research. Although the authors mentioned the aim of this study, new aspect or view point of this research was not clearly stated in the manuscript. Introduction is not well structured. The authors don’t necessarily mention general issues in detail, but are better to show information in a summarized way with focusing on the main issues related to the originality of this study. The authors are strongly encouraged to mention the novel aspects and/or viewpoints which surpass the previous researches in the manuscript clearly.

2) The present Abstract was not informative. Abstract should include purpose of the research, principal results and major conclusions in a summarized way. In addition, due to separation of the Abstract from the major article, it must be a key to lead readers to evoke a spirit of challenge to contact with the contents of the report, as described in Author instructions of this Journal.

3) Another notable aspect is in the experimental methodologies (validations). Although the authors mentioned the Japanese farms that were the subject of this study in Materials and Methods; Sampling locations and sample collection, the type or number of animals kept on the farms, or the specific treatment time (treatment conditions) or amount of treatment related to composting were completely missing in the manuscript. These critically affect the results of this research. The authors are encouraged to take these aspects into account to show the results with enhanced accuracy and reliability of the results.

4) DNA extraction and quantitative PCR (qPCR): Since there is more than one gene region to measure for each of the bla genes (blaTEM, blaCTX-M, and blaSHV) and uidA genes that are the subject of qPCR measurements in this study, specific primer information used for the measurements, with citation of any references, need to be included in the manuscript for the accuracy and better understanding of the results.

5) Statistical processing for results and discussion for the significant difference is desirable for deepen the results and discussions. The authors are encouraged to take these aspects into account to show the results for enhancement of the novelty and better understanding of the results.

6) Resolutions of Figures 1A and 1B, 2, and 3 are not sufficient for publication.

6. PLOS authors have the option to publish the peer review history of their article (what does this mean?). If published, this will include your full peer review and any attached files.

Reviewer #1: No

Reviewer #2: No

---

## [Author Response · Author response to Decision Letter 0]

23 Jan 2024

Author responses to reviewer comments

We are grateful to reviewers 1 and 2 for their insightful comments and suggestions, which have enriched our manuscript. As indicated in the responses below, we have taken all of these comments and suggestions into account in the revised manuscript. Additional and rephrased sentences in the main text are highlighted in yellow.

Comments to the Author

1. Is the manuscript technically sound, and do the data support the conclusions?

Reviewer #1: No

Reviewer #2: Partly

Response: We have rewritten the conclusion.

2. Has the statistical analysis been performed appropriately and rigorously?

Reviewer #1: No

Reviewer #2: N/A

Response: We have performed statistical analysis and described it in the Materials and Methods, and Results sections of the revised manuscript. 

3. Have the authors made all data underlying the findings in their manuscript fully available?

Reviewer #1: No

Reviewer #2: Yes

Response: We have added the raw data in S3 and S4 Tables.

4. Is the manuscript presented in an intelligible fashion and written in standard English?

Reviewer #1: Yes

Reviewer #2: Yes 

5. Review Comments to the Author

Reviewer #1: 1. There are many errors from the viewpoint of science, for example, the description of HGT on Line 60-61, the use of 100 ug/mL for ampicillin resistant experiment on Line 107.

Response: In response to reviewer’s comment, we have rephrased such sentences in the revised manuscript. We used the 100 μg ml-1 ampicillin for the selection of β-lactam-resistant strains, as described previously (reference 17), and rephrased the sentence from “ampicillin-resistant” to “β-lactam-resistant”. 

Line 62-63: Therefore, bla genes are among the important ARGs and should be investigated to determine the risks of dissemination

Line 105-107: without and with 100 μg mL-1 ampicillin (Sigma-Aldrich, St. Louis, MO, USA) to determine the abundance (CFU g-1) of Escherichia coli and coliforms (excluding E. coli), as well as β-lactam-resistant E. coli and coliforms (excluding E. coli), respectively [17].

2. The objective of this work should be more clear elaborated. For example, bla genes as the valid ARG indicator, is it fully convincable?

Response: In response to the reviewer’s comment, we rewrote the corresponding text in the revised manuscript.

Line 62-70: Therefore, bla genes are among the important ARGs and should be investigated to determine the risks of dissemination. To control ARB, the transmission and dynamics of ARGs must be investigated.

If composting is inadequate, residuals containing ARB and ARGs derived from livestock could spread and transmit to soils and crops. This study aimed to investigate the quantitative and qualitative transmission of livestock-derived ARB and ARGs in compost to soils and crops. First, ARB and ARG abundance was determined in cattle feces, composts, agricultural soils, and crops in the same field to evaluate the influence of compost application to agricultural soils. Second, ARB were isolated from cattle feces, composts, agricultural soils, and crops in the same field to clarify the transmission of clonal ARB isolates from livestock to soils through composts.

3. As to the isolation of bacteria, how to differentiate by their colony colors and what does it mean "from samples showing zero E.coli count" on Line 104?

Response: We apologize if our description was unclear. On CHROMagar ECC agar, colonies of E. coli and other coliforms showed blue and red color, respectively. In our study, homogenized samples were inoculated to CHROMagar ECC agar for quantification of E. coli and other coliforms. However, if they did not grow, we conducted a pre-culture step for bacterial isolation. The sentence "from samples showing zero E. coli count" means that colonies of E. coli did not grow on CHROMagar ECC agar.

Line 108-109: according to their colony colors (E. coli, blue colony; other coliforms, red colony)

Line 111: wherein E. coli colonies did not grow,

4. The biggest problem for qPCR is ambitious. Does the sample of isolates (Line 122-123) or feces/compost/soil (Line 128-129) all used? It is rather confusing.

Response: In qPCR, only field collected samples (cattle feces, solid compost, liquid compost, soil, and corn) were used, and isolates were not used. We added the sentence in the revised manuscript.

Line 135: To quantify the genes in each field sample,

5. What does it mean on Line 189-190, Line 195 "bla-positie b-lactam-resistant"?

Response: We apologize if our description was unclear and rephrase the sentence in the revised manuscript.

Line 202-203: E. coli was isolated from all samples using a pre-culture step, excluding the edible parts of corn (day 160) and solid composts.

Line 207-208: The bla genes were not detected in β-lactam-resistant isolates from soil (days 28, 60, 100, and 160), corn, and solid composts.

6. The section of conclusion didn't incorporate with the experimental results and data in this work. It should be improved a lot.

Response: In response to the reviewer’s comment, we have rewritten the Conclusion.

Reviewer #2: On account of the manuscript PONE-D-23-31188, entitled “Low-Frequency Transmission and Persistence of Antimicrobial-Resistant Bacteria and Genes from Livestock to Agricultural Soil and Crops through Compost Application” by Akira Fukuda et al., the authors evaluated the quantitative and qualitative transmission of livestock-derived antimicrobial resistant bacteria (ARB) and antimicrobial resistance genes (ARGs) from livestock to soil and crops through composting in Japan. The topic is important to conduct human health risk assessment for antimicrobial resistance (AMR) in the agricultural environment. After careful consideration, I feel that this manuscript is to be published after improvement of some major shortcomings. Details of my comments are as follows:

1) The view point of this research is interesting, and the authors got interesting results. Several important revisions are, however, required before publication. The first one is novelty of the research. Although the authors mentioned the aim of this study, new aspect or view point of this research was not clearly stated in the manuscript. Introduction is not well structured. The authors don’t necessarily mention general issues in detail, but are better to show information in a summarized way with focusing on the main issues related to the originality of this study. The authors are strongly encouraged to mention the novel aspects and/or viewpoints which surpass the previous researches in the manuscript clearly.

Response: In response to the reviewer’s comment, we have rewritten the Introduction. 

2) The present Abstract was not informative. Abstract should include purpose of the research, principal results and major conclusions in a summarized way. In addition, due to separation of the Abstract from the major article, it must be a key to lead readers to evoke a spirit of challenge to contact with the contents of the report, as described in Author instructions of this Journal.

Response: In response to the reviewer’s comment, we have rewritten the Abstract.

3) Another notable aspect is in the experimental methodologies (validations). Although the authors mentioned the Japanese farms that were the subject of this study in Materials and Methods; Sampling locations and sample collection, the type or number of animals kept on the farms, or the specific treatment time (treatment conditions) or amount of treatment related to composting were completely missing in the manuscript. These critically affect the results of this research. The authors are encouraged to take these aspects into account to show the results with enhanced accuracy and reliability of the results.

Response: In response to the reviewer’s comment, we have added the following texts in the revised manuscript.

Line 75-87: The samples used in this study were collected from the Rakuno Gakuen University farm in Hokkaido, Japan. This encompassed a field spanning 51 000 m2 dedicated to the cultivation of corn. On the field, composts were applied once a year. In the Rakuno Gakuen University farm, approximately 180 dairy cattle were kept, and their excretions were treated via aerobic composting (solid compost) and anaerobic digestion (liquid compost) using a biogas plant. In aerobic composting, the temperature of the solid compost is raised by self-heating. For making a solid compost, 20% dairy cattle excretions, 50% bedding, and 30% feed residues and waste were mixed and stored at a covered, outdoor area. This mixture was turned over every 2 weeks and allowed to stand for 5 months. In anaerobic biogas digestion, liquid compost is treated at a constant temperature, removing most pathogens [16]. For making a liquid compost, 40% dairy cattle excretions, 40 bedding, and 20% farm effluent water were mixed and stored in biogas plant’s tank. In the tank, the mixture was heated to promote fermentation and kept in a month. Solid and liquid composts are kept within their designated areas on the farm. This storage continues until they are ready for use as fertilizer in the soil where corn is cultivated.

4) DNA extraction and quantitative PCR (qPCR): Since there is more than one gene region to measure for each of the bla genes (blaTEM, blaCTX-M, and blaSHV) and uidA genes that are the subject of qPCR measurements in this study, specific primer information used for the measurements, with citation of any references, need to be included in the manuscript for the accuracy and better understanding of the results.

Response: In response to the reviewer’s comment, we listed the specific primers in S2 Table. 

5) Statistical processing for results and discussion for the significant difference is desirable for deepen the results and discussions. The authors are encouraged to take these aspects into account to show the results for enhancement of the novelty and better understanding of the results.

Response: We thank the reviewer for your careful assessment. We have performed statistical analysis and included this information in the Materials and Methods, and Results sections of the revised manuscript. 

6) Resolutions of Figures 1A and 1B, 2, and 3 are not sufficient for publication.

Response: We have reformatted all figures to improve their resolution.

---

## [Decision Letter · Decision Letter 1]

17 Mar 2024

PONE-D-23-31188R1Low-Frequency Transmission and Persistence of Antimicrobial-Resistant Bacteria and Genes from Livestock to Agricultural Soil and Crops through Compost ApplicationPLOS ONE

Dear Dr. Usui,

Thank you for submitting your manuscript to PLOS ONE. After careful consideration, we feel that it has merit but does not fully meet PLOS ONE’s publication criteria as it currently stands. Therefore, we invite you to submit a revised version of the manuscript that addresses the points raised during the review process.

We look forward to receiving your revised manuscript.

Kind regards,

Gabriel Trueba, PhD

Academic Editor

PLOS ONE

Journal Requirements:

Reviewers' comments:

Reviewer's Responses to Questions

**Comments to the Author**

1. If the authors have adequately addressed your comments raised in a previous round of review and you feel that this manuscript is now acceptable for publication, you may indicate that here to bypass the “Comments to the Author” section, enter your conflict of interest statement in the “Confidential to Editor” section, and submit your "Accept" recommendation.

Reviewer #2: All comments have been addressed

Reviewer #3: All comments have been addressed

2. Is the manuscript technically sound, and do the data support the conclusions?

Reviewer #2: Yes

Reviewer #3: Yes

3. Has the statistical analysis been performed appropriately and rigorously? 

Reviewer #2: Yes

Reviewer #3: Yes

4. Have the authors made all data underlying the findings in their manuscript fully available?

Reviewer #2: Yes

Reviewer #3: Yes

5. Is the manuscript presented in an intelligible fashion and written in standard English?

Reviewer #2: Yes

Reviewer #3: Yes

6. Review Comments to the Author

Reviewer #2: On account of the manuscript PONE-D-23-31188R1, entitled “Low-Frequency Transmission and Persistence of Antimicrobial-Resistant Bacteria and Genes from Livestock to Agricultural Soil and Crops through Compost Application” by Akira Fukuda et al., the authors revised the manuscript appropriately according to the Reviewers comments. After careful consideration, I made a decision that the manuscript is acceptable for publication in its present form.

Reviewer #3: Dear Authors

The manuscript "Low-Frequency Transmission and Persistence of Antimicrobial-Resistant Bacteria and Genes from Livestock to Agricultural Soil and Crops through Compost Application" is an interesting study.

1. However, the "Abstract" section does not contain any of the numerical/ quantitative results of the study. It is suggested to include these results.

2. Critical and speculative statement needs to be incorporated in discussion regarding "why or how the burden is decrease after 7 days in soil/ compost".

Thanks and Regards

7. PLOS authors have the option to publish the peer review history of their article (what does this mean?). If published, this will include your full peer review and any attached files.

Reviewer #2: No

Reviewer #3: **Yes: **Muhammad Asif Zahoor

---

## [Author Response · Author response to Decision Letter 1]

18 Mar 2024

Author responses to reviewer comments

We are grateful to reviewers 1 and 2 for their insightful comments and suggestions, which have enriched our manuscript. As indicated in the responses below, we have taken all of these comments and suggestions into account in the revised manuscript. Additional and rephrased sentences in the main text are highlighted in yellow.

Reviewer #2

1. However, the "Abstract" section does not contain any of the numerical/ quantitative results of the study. It is suggested to include these results.

Response: In response to the reviewer’s comment, we have added the following texts in the revised manuscript.

Line 26-27: In the soil, 5.0 CFU g-1 E. coli was detected on day 0 (the day post-compost application), and then, E. coli was not quantified on 60 days post-application.

2. Critical and speculative statement needs to be incorporated in discussion regarding "why or how the burden is decrease after 7 days in soil/ compost".

Response: In response to the reviewer’s comment, we have added the following texts in the revised manuscript.

Line 245-249: Agricultural soils would be a suitable environment for fecal bacteria [25]. In the absence of antimicrobial selective pressure, it is diﬃcult for ARB to maintain dominance among the total microbes of an environment because plasmids carrying ARGs require ﬁtness costs and this causes inferior growth of ARB compared with that of the wild-type strains [30].

---

## [Editor Report · Decision Letter 2]

27 Mar 2024

Low-Frequency Transmission and Persistence of Antimicrobial-Resistant Bacteria and Genes from Livestock to Agricultural Soil and Crops through Compost Application

PONE-D-23-31188R2

Dear Dr. Usui,

We’re pleased to inform you that your manuscript has been judged scientifically suitable for publication and will be formally accepted for publication once it meets all outstanding technical requirements.

Kind regards,

Gabriel Trueba, PhD

Academic Editor

PLOS ONE
---

## [Editor Report · Acceptance letter]

10 May 2024

PONE-D-23-31188R2 

PLOS ONE

Dear Dr. Usui, 

I'm pleased to inform you that your manuscript has been deemed suitable for publication in PLOS ONE. Congratulations! Your manuscript is now being handed over to our production team.

Kind regards, 

on behalf of

Dr. Gabriel Trueba 

Academic Editor

PLOS ONE